# Cost-effectiveness of a patient-centred approach to managing multimorbidity in primary care: a pragmatic cluster randomised controlled trial

Joanna Thorn [1], Mei-See Man [1,2] Katherine Chaplin [1], Peter Bower,[3] Sara Brookes,[2] Daisy Gaunt,[2] Bridie Fitzpatrick,[4] Caroline Gardner,[3] Bruce Guthrie,[5] Sandra Hollinghurst,[1] Victoria Lee,[3] Stewart W Mercer,[4] Chris Salisbury [1]

For numbered affiliations see end of article.

**Correspondence to**
Dr Joanna Thorn;
joanna.thorn@bristol.ac.uk

## ABSTRACT

**Objective** Patients with multiple chronic health conditions are often managed in a disjointed fashion in primary care, with annual review clinic appointments offered separately for each condition. This study aimed to determine the cost-effectiveness of the 3D intervention, which was developed to improve the system of care.

**Design** Economic evaluation conducted alongside a pragmatic cluster-randomised trial.

**Setting** General practices in three centres in England and Scotland.

**Participants** 797 adults with three or more chronic conditions were randomised to the 3D intervention, while 749 participants were randomised to receive usual care.

**Intervention** The 3D approach: comprehensive 6-monthly general practitioner consultations, supported by medication reviews and nurse appointments.

**Primary and secondary outcome measures** The primary economic evaluation assessed the cost per quality-adjusted life year (QALY) gained from the perspective of the National Health Service (NHS) and personal social services (PSS). Costs were related to changes in a range of secondary outcomes (QALYs accrued by both participants and carers, and deaths) in a cost–consequences analysis from the perspectives of the NHS/PSS, patients/carers and productivity losses.

**Results** Very small increases were found in both QALYs (adjusted mean difference 0.007 (−0.009 to 0.023)) and costs (adjusted mean difference £126 (£−739 to £991)) in the intervention arm compared with usual care after 15 months. The incremental cost-effectiveness ratio was £18 499, with a 50.8% chance of being cost-effective at a willingness-to-pay threshold of £20 000 per QALY (55.8% at £30 000 per QALY).

**Conclusions** The small differences in costs and outcomes were consistent with chance, and the uncertainty was substantial; therefore, the evidence for the cost-effectiveness of the 3D approach from the NHS/PSS perspective should be considered equivocal.

**Trial registration number** ISCRTN06180958

## INTRODUCTION

The number of patients living with multiple chronic health conditions (multimorbidity)

### Strengths and limitations of this study

► This economic evaluation was conducted alongside the largest randomised controlled trial of an intervention for managing multimorbid patients in primary care.
► Data collection was meticulous and high questionnaire return rates were achieved.
► Data on the use of care homes, which may contribute significantly to social care costs, were not available to include in the economic evaluation.
► Estimates of healthcare costs for this type of intervention have wide CIs, even with a substantial sample size.

is increasing in developed countries as the population ages.[1] Multimorbidity is associated with poor health-related quality of life,[2 3] higher use of health services and higher costs.[4 5] Multimorbidity is common for older adults (with a prevalence of up to 98%),[4] leading to a substantial economic burden.[6]

Care pathways for patients with multimorbidity are often poorly coordinated and burdensome for patients. Each condition may be treated separately, which fails to take account of interactions between diseases.[1] There have been relatively few randomised controlled trials (RCTs) of interventions designed to address issues affecting multimorbid patients, with mixed findings leading to substantial uncertainty about the effectiveness of such interventions.[7] Evidence on the cost-effectiveness of interventions remains even more limited and the evidence that is available is inconsistent.[7–9] A recent review of comprehensive care programmes for multimorbidity found no evidence that such interventions reduce healthcare costs or primary care visits.[10] Several reviews have highlighted the need for research on new interventions,

including assessment of cost-effectiveness.[1 7 11] The inconsistency in the evidence for cost-effectiveness provided a clear indication that a large, well-conducted trial with an integral economic evaluation needed to be carried out.

The '3D' intervention was developed to address the issues associated with managing patients with multimorbidity in primary care in the UK. The intervention was based on a patient-centred care model and aimed to improve continuity of care with caregivers in general practitioner (GP) surgeries, reduce patient burden in accessing healthcare and increase patient involvement in decision-making about their care (see box 1). Based on the seminal work of Stewart,[12] the concept of patient-centred care includes a focus on the patient's individual illness experience, an integrated biopsychosocial perspective, seeking to find common ground with the patient and agreeing management plans and enhancing a continuing relationship between the patient and doctor. We sought to apply these principles within the 3D intervention. The 3D randomised trial assessed the effectiveness of the 3D

approach compared with usual care, and incorporated an embedded process evaluation. The trial did not find any differences in health-related quality of life over 15 months of follow-up, but the intervention did lead to improved patient-centred care.[13 14] Here, we report the results of the economic evaluation conducted alongside the 3D trial.

## METHODS
### Design: the 3D trial
Full details of the study design, eligibility criteria, recruitment process and clinical effectiveness results are given elsewhere.[14 15] All patients and practices gave written informed consent to participate in the study. In brief, the pragmatical 3D cluster randomised trial compared the effectiveness and cost-effectiveness of a complex intervention for multimorbidity with usual care delivered in 33 general practices in Scotland and England. The target patients were adults with multimorbidity, defined as having three or more chronic conditions from those included in the National Health Service (NHS) Quality and Outcomes Framework.[16] Clinical staff in intervention practices attended two half-day 3D training sessions. Participants in the intervention arm were offered 6-monthly holistic reviews of their health problems. Each review consisted of a nurse review appointment to collect relevant clinical information, a medication review conducted by a pharmacist and an extended appointment for the patient to discuss their conditions with a named GP. An individual health plan was negotiated with patients, detailing their own priorities for managing their conditions. The primary outcome measure for effectiveness was health-related quality of life at 15 months after randomisation, measured using the EQ-5D-5L.[17]

The primary economic cost–utility analysis was conducted from the NHS and personal social services (PSS) perspective over 15 months of follow-up. A secondary analysis was conducted from the perspective of patients and carers.

### Patient and public involvement
An active group of up to 14 patients and carers provided a service user perspective, contributing to refinement of the research questions, design of the intervention, design of outcome measures, analysis of qualitative data, patient newsletters, the study website and interpretation of findings. All participants were given a summary of the findings and a link to the published paper describing the main results. One of the outcome measures in the study was treatment burden and this was compared between trial arms.

### Measures
#### Outcome measurement
The primary economic outcome measure was quality-adjusted life years (QALYs). EQ-5D-5L measurements were collected by postal questionnaires at baseline, 9

---

> **Box 1 The 3D model and usual care in the UK**
>
> **The 3D model**
> 3D is a name used to allude to the concept of wholeness, while also acting as a mnemonic for 'dimensions of health, depression and drugs'.
> ► Based on a patient-centred care model.
> ► Practice level changes to improve continuity of care and to replace disease-focused reviews of each health condition with one 6 monthly comprehensive 3D review with a named GP.
> ► Each 3D review consisted of three elements;
>    1. Nurse consultation to identify health problems most important to the patient, issues with quality of life, screening for depression, collecting health data, for example, blood pressure and information relevant to the patient's specific conditions.
>    2. Pharmacist review of medication from medical records, aiming to simplify and optimise drug treatment. Pharmacists were asked to identify non-essential drugs that could be stopped and essential drugs that should be started, and to seek ways to simplify drug treatment regimes, for example, by making all doses once daily.
>    3. GP reviewed data from nurse and pharmacist, and agreed a health plan with the patient, which was given to them as a printed copy.
> ► An interactive computerised template enabled sharing of data between the clinicians, tailored the review to the patient's specific conditions, contained automated checks and generated printed summaries to enable sharing of information and plans with the patient.
> ► Strategies to encourage implementation were used, including training, monthly feedback about implementation and financial incentives for completed reviews.
>
> **Usual care in the UK**
> ► Review of chronic conditions is mainly carried out by nurses in primary care using disease-specific data-entry screens or templates.
> ► Nurses commonly specialise in particular conditions and review each disease separately.
> ► Chronic disease reviews mainly focus on meeting the requirements of the Quality and Outcomes Framework (QOF) pay-for-performance scheme.

months and 15 months post-recruitment, supplemented by telephone collection for non-responders. Utility scores based on a UK population were derived from responses to the EQ-5D-5L cross-mapped to valuations obtained for the EQ-5D-3L instrument.[18] This was a change (approved by the Independent Data Monitoring Committee) to the analysis originally planned, as the National Institute for Health and Care Excellence (NICE) issued a position statement recommending this approach[19] subsequent to the design of the study but prior to analysis. QALYs over the 15-month period were formed from these valuations by means of linear interpolation and an area under the curve calculation. Patients who died during follow-up were treated as if their most recent utility score was relevant until death, and set to zero immediately at death. Carers also completed the EQ-5D-5L instrument at the same timepoints as patients.

### Resource-use measurement

As the trial population had multimorbidity, resource use related to any health condition experienced by the participant was considered relevant.

Staff training attendance records were kept to track resources used to deliver the training programmes for GPs, nurses and receptionists, including trainee and trainer delivery time (and preparation time), travel costs and course materials. 3D GP appointments, nurse appointments and pharmacist reviews were captured by downloading routine GP records, supplemented by manual data capture by researchers reviewing participants' medical records at the end of the trial.

Data on medications prescribed and tests/investigations in primary care were extracted from downloads of routine GP records. Details of the number and duration of primary care consultations (other than 3D reviews) were similarly extracted. These included face-to-face, telephone and home consultations with doctors, nurses or healthcare assistants based in general practice. Duration details were not available for practices in Scotland. Therefore, an average duration for each type of consultation by each staff type in each arm was derived using available data (practices in England only) and applied to all relevant consultations. The research team collected NHS secondary care data from participants' GP records.

To assess travel costs, the patient's normal transport method for GP appointments and the cost (for public transport) or mileage (for private transport) were collected in the patient-reported questionnaire at baseline, along with usual practice with regards to time off work, and details of whether the participant paid prescription charges or not. NHS community care, care from social services, time off work for patients and carers to attend hospital appointments and over-the-counter medication expenditure were captured in the patient-reported questionnaires at 9 and 15 months, as were private use of treatments and therapies.

### Valuation of resource use

Unit costs for NHS staff time for training and delivery of the intervention were based on the most recently available national estimates.[20] Actual expenses relating to training materials, refreshments and staff travel were recorded. Because only some GPs and a small proportion of patients in each practice participated in the trial, costs were adjusted to estimate the cost of training a full practice, divided by the number of eligible patients in the practice and annualised over an estimated 5-year period of relevance. Medication costs were downloaded directly as part of the routine GP records, supplemented by estimates from the British National Formulary where necessary.[21] Patient prescription charges were subtracted from the NHS perspective medication costs. Community and primary care costs were based on recent national estimates.[20]

Secondary care contacts were assigned to a healthcare resource group (groups of events that have been judged to consume similar levels of resources) and costed based on national reference costs.[22] Productivity costs for both patients and carers were estimated based on average weekly earnings stratified by age group.[23] Mileage costs were estimated using UK government allowances.[24] Costs for over-the-counter medication and private therapies and treatments were all reported directly by patients. Unit costs used in the analyses are detailed in online supplementary appendix 1.[14]

All costs were reported in 2015/2016 pounds sterling, adjusted for inflation where necessary.[25] Costs and outcomes occurring during the final 3 months of follow-up were discounted in line with NICE guidance (currently 3.5%).[26] Dates were not available for all types of resource use measured in the trial; in these cases, 50% of the costs incurred in the final 6 months of follow-up were subjected to discounting.

### Data cleaning and missing costs and outcomes

Data cleaning was undertaken prior to unblinding. Questionnaires were not classed as 'missing data' for the cost analysis unless the questionnaire was not returned or the majority of responses were uninterpretable. Medication costs downloaded from practices were manually amended where they were clearly wrong on visual inspection (eg, a prescription for a salbutamol inhaler with a recorded cost of over £1000). The primary analysis included all participants using imputation to predict missing costs and outcomes.[27] Data imputed (using chained equation multiple imputation methods) for the main statistical analysis were used for the economic evaluation.[13 14 28 29] To facilitate convergence of the imputation model, costs were imputed using aggregated cost categories (medications, pharmacy reviews, secondary care, primary care, social care and other types of care) rather than at the level of individual resource-use items.

### Economic analyses

Statistical analyses were conducted using Stata 14.2.[30] All analyses were by 'intention to treat' (comparing the two

groups as randomised and including all patients in the primary analysis). A glossary of economic terms is given in online supplementary appendix 2, and more details may be found in Morris *et al*.[31]

A cost–utility analysis was conducted from the NHS/PSS perspective corresponding to the NICE reference case.[26] The incremental mean difference in QALYs between the two arms of the trial and 95% CIs were derived. Overall mean NHS/PSS costs, and SEs for both arms of the trial were calculated. The incremental mean difference in total costs between the two arms of the trial and 95% CIs were estimated.

Cost and QALY data were combined to calculate an incremental cost-effectiveness ratio (ICER) and net monetary benefit (NMB) statistic[32] from the NHS/PSS perspective. In the primary analysis we estimated whether the 3D approach was cost-effective at the established NICE thresholds of £20 000 and £30 000 per QALY gained. The probability that the 3D approach was cost-effective at various societal 'willingness to pay for a QALY' thresholds was depicted using a cost-effectiveness acceptability curve (CEAC). All measures of cost-effectiveness (ICER, CEAC and NMB) and CIs were derived parametrically using the output of seemingly-unrelated regression analysis to account for the correlation between costs and outcomes,

and controlling for baseline imbalance in utility for the QALY equation[33]. Clustering within GP practices was accounted for by including the randomisation variables in the regression.

As the broader societal perspective also has relevance, the costs of each component of the intervention were estimated separately from the NHS/PSS perspective (ie, including all resources supplied by the healthcare provider), the patient/carer perspective (ie, including all costs borne by the participants themselves) and the productivity perspective (ie, taking into account time off work) and related to changes in a range of secondary outcomes in a cost–consequences analysis.[34] Consequences included QALYs accrued by both participants and carers, and deaths. The cost–consequences analysis was based on available cases, which differed in number for each type of healthcare resource or outcome; an available case was defined as an individual having complete data for each relevant timepoint. Linear regression output was used to derive CIs parametrically, accounting for clustering within practices.

### Sensitivity analyses

One-way sensitivity analyses were used to judge the potential impact of sources of uncertainty. This included an

| Table 1 | Baseline characteristics of practices and patients | |
|---|---|---|
| | **Usual care**<br>**(n=17 practices, 749 patients)** | **Intervention**<br>**(n=16 practices, 797 patients)** |
| Patients | | |
| Age: mean (SD) | 70.7 (11.4) | 71.0 (11.6) |
| Female: no. (%) | 377 (50%) | 406 (51%) |
| White ethnicity: no./total no. (%) | 729/739 (99%) | 775/780 (99%) |
| Fully retired from work: no./total no. (%) | 512/721 (71%) | 525/759 (69%) |
| No. of long-term conditions from QOF: median (IQR) | 3.0 (3.0 to 3.0) | 3.0 (3.0 to 3.0) |
| No. of self-reported conditions: median (IQR), n | 7.0 (5.0 to 10.0), 748 | 7.0 (5.0 to 9.0), 795 |
| Long-term conditions* no. (%) with: | | |
| Cardiovascular disease or chronic kidney disease† | 698 (93%) | 747 (94%) |
| Stroke or transient ischaemic attack | 241 (32%) | 286 (36%) |
| Diabetes | 401 (54%) | 411 (52%) |
| Chronic obstructive pulmonary disease or asthma | 382 (51%) | 388 (49%) |
| Epilepsy | 35 (5%) | 41 (5%) |
| Atrial fibrillation | 249 (33%) | 281 (35%) |
| Serious mental illness‡ | 37 (5%) | 29 (4%) |
| Depression | 283 (38%) | 276 (35%) |
| Dementia | 27 (4%) | 33 (4%) |
| Learning disability | 7 (1%) | 7 (1%) |
| Rheumatoid arthritis | 55 (7%) | 48 (6%) |
| EQ-5D-5L score: mean (SD), n | 0.542 (0.292), 747 | 0.574 (0.282), 795 |

*Conditions with similar clinical management were grouped and counted once only.
†Including coronary heart disease, hypertension, heart failure, peripheral arterial disease, chronic kidney disease stages 3 to 5.
‡Including schizophrenia, psychosis, bipolar disease.
QOF, Quality and Outcomes Framework.

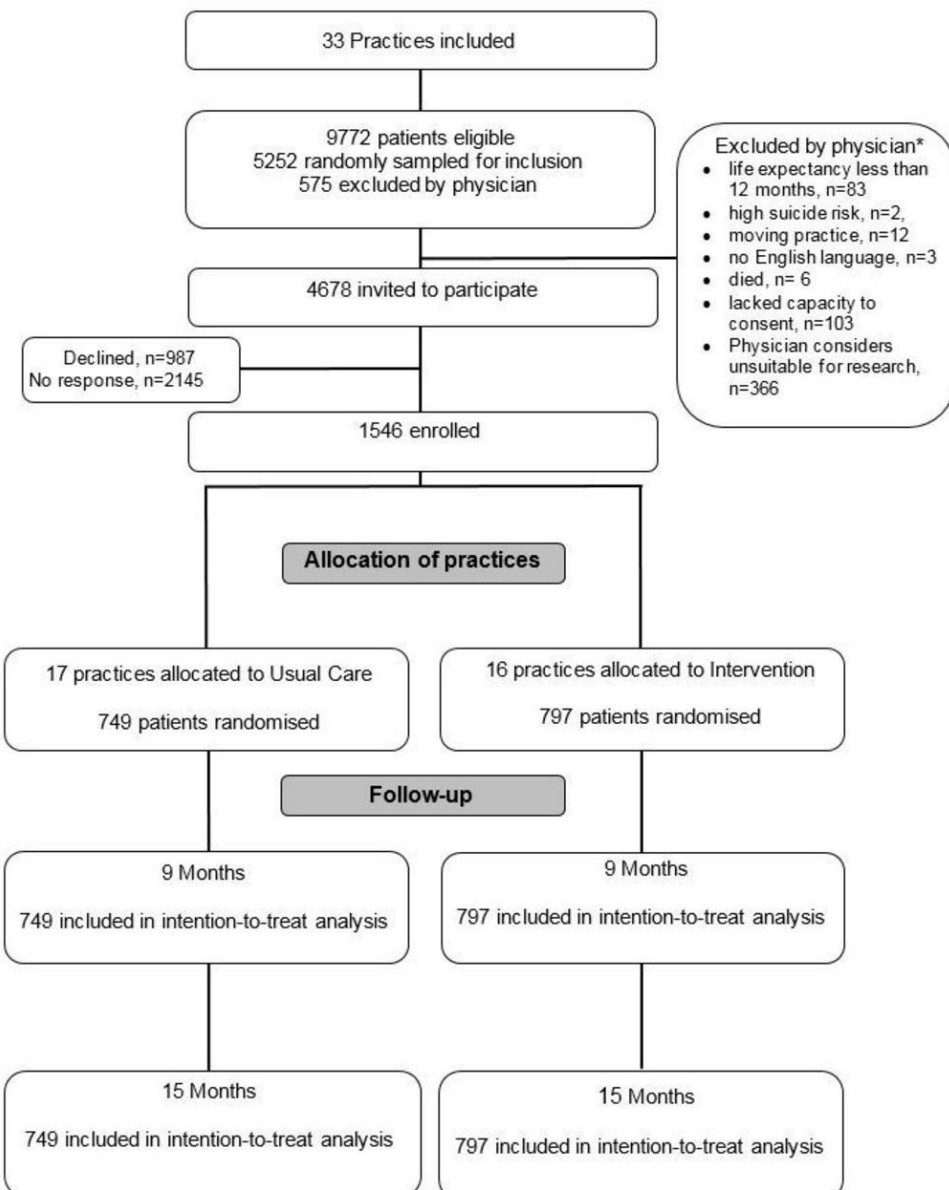

**Figure 1** Trial profile † † Salisbury C, man M-S, Bower P, *et al*. Management of multimorbidity using a patient-centred care model: A pragmatic cluster-randomised trial of the 3D approach. *The Lancet* 2018;392(10141):41–50. *categorised according to exclusion criteria.

analysis excluding participants who died to assess the impact of the imbalance in deaths between arms, and an analysis without discounting either costs or outcomes (ie, a discount rate of zero) to assess the impact of the discount rate. A complete-case analysis was also conducted to assess the impact of the imputation process; a complete case was defined as a participant for whom full resource-use data and full outcome data were available.

## RESULTS

The results are reported according to the specifications of the Consolidated Health Economic Evaluation Reporting Standards statement.[35] Baseline characteristics of the practices and patients are given in table 1,[13] and the flow of patients through the study is illustrated in figure 1.[13]

### Missing data

Sixteen practices (797 participants) were randomised to be offered the 3D approach, while 17 practices (749 participants) were randomised to receive usual care. Missing data occurred for a number of reasons, including withdrawal from the trial, or leaving the practice. Twelve participants (1.5%) in the 3D arm and six (0.8%) in the control arm had no information on secondary care use because it was not possible to locate their medical records (p=0.2). Practice downloads of medication and investigation data failed for 18 participants (2.3%) in the intervention arm and eight (1.1%) in the usual care arm (p=0.07), while 19 (2.4%) and 10 (1.3%) 3D and usual care participants, respectively, were missing consultation data from practice downloads (p=0.13). Not all participants

returned all questionnaires at all timepoints. 165 participants (20.7%) in the intervention arm and 125 (16.7%) in the control arm did not return a questionnaire at one or more follow-up points (p=0.04). Not all those who did return questionnaires completed the resource-use questions; in total, 181 (22.7%) in the intervention arm and 146 (19.5%) in the control arm were missing resource-use data from questionnaires at one or more follow-up points (p=0.12). Complete data sets were available for 1191 participants in total (599 (75.2%) in the 3D arm and 592 (79%) in the control arm, p=0.07). Participants with missing data were in a significantly poorer health state at baseline (mean (95% CI) EQ-5D-5L score: 0.453 (0.422 to 0.485)) than participants with full data sets (mean (95% CI) EQ-5D-5L score: 0.589 (0.574 to 0.605)).

### Primary analysis
#### Outcomes and resource use
The primary analysis using imputed data showed that participants in the intervention arm gained a mean of 0.007 (95% CI: −0.009 to 0.023) additional QALYs over 15 months compared with participants in the usual care arm. Total costs per patient from the NHS/PSS perspective were £126 (95% CI: -£739 to £991) higher in the intervention arm than in the usual care arm. Wide variability was observed in both costs and outcomes, with both CIs overlapping zero. Disaggregated resource-use data are in online supplementary appendix 3.[14]

#### Cost-effectiveness of 3D
Cost-effectiveness statistics from the NHS/PSS perspective are given in table 2.[14] The ICER was £18 499, and the net monetary benefit at a societal willingness-to-pay value of £20 000 was £10 (95% CI: -£956 to £977). At this willingness-to-pay value, the probability that the 3D

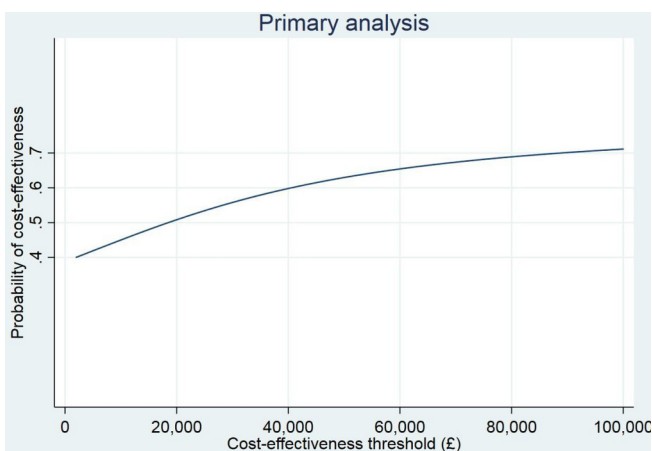

**Figure 2** Cost-effectiveness acceptability curve from the NHS/PSS perspective.† † Salisbury C, man M-S, Chaplin K, Mann C, Bower P, Brookes S, *et al*. A patient-centred intervention to improve the management of multimorbidity in general practice: The 3D RCT. Health Serv Deliv Res 2019;7(5). NHS, National Health Service; PSS, personal socialservices; RCT, randomised controlled trials.

approach is cost-effective was 0.508, while at £30 000, the probability of cost-effectiveness was 0.558. A cost-effectiveness acceptability curve depicting the probability of cost-effectiveness at a range of willingness-to-pay values is given in figure 2.[14]

### Sensitivity analyses
Results from an analysis restricted to complete cases only are given in table 3.[14] The complete-case analysis suggested that the 3D approach was associated with both lower costs and better outcomes, with a probability of cost-effectiveness of 0.705 at a willingness-to-pay threshold of £20 000. A sensitivity analysis excluding participants who

| Table 2 Cost-effectiveness of the 3D approach from an NHS and personal social services perspective | | | |
|---|---|---|---|
| | **Usual care mean (SE) n=749** | **Intervention mean (SE) n=797** | **Incremental difference (95% CI)** |
| **Costs (£)** | | | |
| Unadjusted costs from the NHS/PSS perspective | 6032 (362) | 6124 (317) | |
| Adjusted costs from the NHS/PSS perspective | 6014 (343) | 6140 (333) | 126 (−739 to 991) |
| **Outcomes** | | | |
| Unadjusted QALYs over 15 months of follow-up | 0.651 (0.013) | 0.691 (0.012) | |
| Adjusted QALYs over 15 months of follow-up | 0.668 (0.006) | 0.675 (0.006) | 0.007 (−0.009 to 0.023) |
| **Cost-effectiveness statistics** | | | |
| ICER: £18 499 | | | |
| Net monetary benefit at £20 000 (95% CI): £10 (−956 to 977) | | | |

Net monetary benefit at £30 000 (95% CI): £78 (−974 to 1130).
Cost estimates were adjusted for randomisation variables; QALY estimates were adjusted for randomisation variables and baseline utility.
CI, confidence interval; ICER, incremental cost-effectiveness ratio; NHS, National Health Service; PSS, personal social services; QALY, quality-adjusted life year; SE, standard error.

**Table 3** Sensitivity analysis: cost-effectiveness of the 3D approach from an NHS/PSS perspective based on complete cases only

| | Usual care mean (SE) n=592 complete cases | Intervention mean (SE) n=599 complete cases | Incremental difference (95% CI) |
|---|---|---|---|
| **Costs (£)** | | | |
| Unadjusted costs from the NHS/PSS perspective | 4916 (290) | 4757 (222) | |
| Adjusted costs from the NHS/PSS perspective | 4905 (258) | 4768 (256) | −137 (−852 to 577) |
| Outcomes | | | |
| Unadjusted QALYs over 15 months of follow-up | 0.698 (0.014) | 0.750 (0.013) | |
| Adjusted QALYs over 15 months of follow-up | 0.722 (0.005) | 0.726 (0.005) | 0.004 (−0.010 to 0.019) |
| **Cost-effectiveness statistics** | | | |
| ICER: Intervention dominates | | | |
| Net monetary benefit at £20,000: £222 (−584 to 1028). | | | |

Cost estimates were adjusted for randomisation variables; QALY estimates were adjusted for randomisation variables and baseline utility.
CI, confidence interval; ICER, incremental cost-effectiveness ratio; NHS, National Health Service; PSS, personal social services; QALY, quality-adjusted life year; SE, standard error.

died suggested that the probability of cost-effectiveness of the 3D approach at £20 000 was 0.561. A further sensitivity analysis using undiscounted costs and outcomes did not suggest that the discount rate affected the conclusions.

### Cost–consequences analysis

Costs and selected outcomes (on an available case basis) are presented in table 4[14] from the primary perspective of the NHS/PSS and the secondary perspective of the patient/carer themselves alongside an estimate of the societal loss of productivity.

Costs from all perspectives were very similar between arms and no cost group differed significantly (other than those associated with the intervention itself). Other than for day case/outpatient care, emergency care and medications, costs to the NHS were higher in the intervention arm than in the usual care arm, while social services usage was higher in the usual care arm. Overall costs (based on available data) from the NHS/PSS perspective were slightly higher in the usual care arm although the difference was consistent with a chance finding.

Costs borne by patients and carers were higher overall in the intervention arm, although the medication costs (both prescription charges and over-the-counter remedies) were slightly lower; no patient cost group exhibited a statistically significant difference. The cost of productivity losses was similar in the two arms (and statistically consistent with chance), although slightly higher in the 3D approach arm.

QALYs (adjusted for baseline utility scores) were slightly higher for patients and lower for carers in the intervention arm than in the usual care arm; however, the difference was consistent with chance. Although there was a higher number of deaths in the intervention arm, the difference was not statistically significant.

### DISCUSSION
#### Principal findings

No significant difference was observed between participants offered the 3D intervention and those receiving usual care only for overall costs, resource use of any category or QALY outcomes. The participants were substantial users of healthcare, with inpatient hospital care and medications both high contributors to overall costs. A small improvement in outcome (representing an additional 2.6 days of the best imaginable health over 15 months) coupled with a small increase in costs between arms led to weak evidence for the cost-effectiveness of the 3D approach in the primary analysis from the NHS/PSS perspective. The net monetary benefit was small, but positive, indicating that society is willing to pay for the benefits that can be achieved.

#### Strengths and limitations

This economic evaluation was conducted alongside a large and rigorous RCT. Meticulous data collection practices allowed individual patient data to be measured for all the key cost drivers, a particularly challenging task for a patient group that has high healthcare usage. The study contributes to the body of evidence supporting the care of patients with multimorbidity. Patterns of missing resource-use data were similar between arms, and high questionnaire return rates were achieved[36]; however, the recall period for resource-use was quite long at 9 months, which may have led to patients omitting to include some

**Table 4**  Costs and consequences of the 3D approach and usual care

| Costs and outcomes | Usual care | N | Intervention | N | Difference (95% CI) |
|---|---|---|---|---|---|
| Mean costs from the NHS perspective (£) | | | | | |
| Practice-based consultations | 627 | 754 | 726 | 715 | 99 (−7 to 205) |
| Practice-based investigations | 45 | 755 | 61 | 717 | 15 (−6 to 37) |
| Community-based healthcare | 160 | 615 | 167 | 601 | 7 (−35 to 49) |
| Inpatient stays | 1867 | 766 | 1920 | 722 | 52 (−470 to 574) |
| Outpatient visits and day cases | 614 | 766 | 613 | 722 | −1 (−168 to 167) |
| Accident and emergency visits | 102 | 766 | 99 | 722 | −3 (−24 to 19) |
| Ambulance trips to hospital | 131 | 615 | 141 | 601 | 10 (−56 to 77) |
| Prescribed medications | 1230 | 755 | 1221 | 717 | −8 (−220 to 203) |
| Pharmacy reviews | 0 | 766 | 8 | 722 | 8 (7 to 9) |
| Intervention set up | 0 | 797 | 4 | 749 | 4 (3 to 5) |
| Social services | 559 | 615 | 403 | 601 | −156 (−476 to 164) |
| All NHS/PSS | 4929 | 609 | 4746 | 598 | −183 (−923 to 556) |
| Mean costs from the patient/carer perspective (£) | | | | | |
| Prescription charges | 5 | 755 | 4 | 717 | −2 (−6 to 2) |
| Travel to GP practice | 24 | 749 | 34 | 711 | 10 (−4 to 24) |
| Over-the-counter medications | 39 | 615 | 35 | 601 | −3 (−16 to 9) |
| Private healthcare | 93 | 615 | 122 | 601 | 29 (−40 to 97) |
| All patient/carer | 162 | 608 | 195 | 597 | 33 (−35 to 101) |
| Mean productivity loss (£) | | | | | |
| Productivity loss | 122 | 608 | 161 | 597 | 39 (−47 to 125) |
| Outcomes | | | | | |
| QALYs (patient) | 0.693 | 665 | 0.695 | 647 | 0.003 (−0.013 to 0.019) |
| QALYs (carer) | 0.943 | 41 | 0.920 | 50 | −0.024 (−0.064 to 0.017) |
| Deaths | 32 | 749 | 46 | 797 | p=0.18 ($\chi^2$ test) |

All costs and consequences are based on available data; the totals from each perspective are not, therefore, equal to the sum of the components. CI were calculated using SEs from standard linear regressions adjusted for cluster at the level of the practice. QALYs were adjusted for baseline utility scores.

CI, confidence interval; GP, general practitioner; NHS, National Health Service; PSS, personal social services; QALY, quality-adjusted life year.

resources. Medication costs were based on prescriptions issued by the GP, and it is not certain that all scripts were filled by the participant; the medication costs may, therefore, be overestimated. Although the medication data were checked carefully, the volume of medications prescribed to the 3D population rendered it infeasible to check every entry and some errors may have persisted.

Use of care homes, which can be a significant contributor to costs of social care, was not included in the economic evaluation. The funding of care homes within the UK is complex, with patients often paying considerable amounts themselves. However, only 13 participants reported living in a care home and this did not differ by arm. The follow-up period of the 3D trial was 15 months, and longer-term outcomes are unknown; a follow-up period of 2 to 3 years might have allowed an effect on quality of life or healthcare resource use to be observed. Use of simple mean imputation methods for estimating missing information (such as the cost of a bus fare) in the

questionnaire data will have reduced SEs and underestimated the uncertainty around these costs.

### Comparison with other studies

Comparison with other studies is hampered by the variety of definitions used for multimorbidity, and by the heterogeneity in study designs.[11] Evidence concerning the cost-effectiveness of interventions designed to manage patients with multimorbidity is limited; only two of the RCTs identified in the review by Smith et al undertook cost-effectiveness analyses.[7] Katon et al reported that implementation of a collaborative care intervention for patients with depression alongside either diabetes or coronary heart disease was justified on cost-effectiveness grounds.[37] However, the authors estimated QALYs based on clinical outcomes rather than by direct measurement, and there was substantial uncertainty around both costs and outcomes. Barley et al found that a nurse-led patient-centred intervention was cost-effective up to a willingness-to-pay threshold of approximately £3000 per QALY; however, the study was only a pilot RCT, and

multimorbidity was defined to include patients with coronary heart disease and depression only.[38] In a feasibility trial, Mercer *et al* found that a whole-system intervention designed for patients with two or more chronic conditions in deprived areas may be considered cost-effective.[39] There is a clear need for further well conducted studies on the cost-effectiveness of interventions for multimorbid patients.

### Implications of study

This is the largest trial of an intervention for multimorbidity ever undertaken; however, there remains substantial uncertainty surrounding the results. The impact of organisational changes in delivery of primary healthcare on patients' quality of life is usually very small and the costs are also usually low, but skewed by a small number of patients with high costs in secondary care often unrelated to the intervention. This leads to wide CIs that cross zero (ie, substantial uncertainty) for estimates of cost-effectiveness and illustrates the difficulty of obtaining precise estimates for the economic outcomes of this type of intervention even in large trials.

The 3D participants had a mean(SD) utility of 0.558 (0.287) at entry to the study, which compares poorly to a UK population mean of 0.779 for ages 65 to 74.[40] Participants had a small positive increase in QALYs in the intervention arm, while carers had a small decrease in QALYs compared with the usual care arm; it is possible that an analysis taking carer outcomes into account might reach a different conclusion. The small contribution to overall costs made by productivity losses is consistent with the predominantly retired study population; at baseline, over 65% of participants described themselves as 'fully retired from work'.

Anticipated reductions in appointment attendances were not achieved, partly because some patients attended 3D reviews in addition to, rather than instead of, their single-condition reviews. Similarly, although cost savings could have been achieved through a reduction in prescriptions issued, this was not observed in the trial; although the brief was to seek to simplify patients' medication regimes, the embedded process evaluation[14] suggested that pharmacists appeared to concentrate on adherence to guidelines rather than whether the patient might benefit from stopping some medications. There was substantial variation between practices in the extent to which they implemented the intervention and this may have contributed to variation in costs in the trial arm. Three practices barely implemented the intervention at all because of practice or staffing problems unconnected with the intervention, and this will have diluted the potential effects of the intervention overall. It is possible that the cost-effectiveness of the intervention could be increased by focusing more on patients with the greatest needs, by more attention to de-prescribing and by ending unnecessary additional disease-specific reviews as the 3D model became normalised.

### Future work

The EQ-5D instrument was used because it is recommended by NICE in the UK,[26] and there was little evidence on alternatives at the trial outset. However, it is difficult to pick up small changes in health-related quality of life arising from service reconfigurations, and it may be that a measure capturing outcomes beyond health would be more appropriate. In future work, researchers should consider including alternative economic outcome measures alongside the EQ-5D; for example, the ICECAP suite of capability well-being instruments.[41] The effect of the intervention on carers is being considered further, and it would be interesting to explore the reasons for the errors observed in the medication records downloaded from GP practice systems.

### CONCLUSIONS

The evidence for the cost-effectiveness of the 3D intervention is equivocal; the results suggest that there is just over a 50% chance of cost-effectiveness at the established threshold of £20 000 per QALY from the NHS/PSS perspective. The small differences in costs and outcomes are consistent with chance, and the uncertainty is substantial; therefore, the results should be interpreted with caution. Given the equivocal nature of the cost-effectiveness results, implementation of the intervention cannot be recommended on economic grounds alone.

**Author affiliations**

[1]Centre for Academic Primary Care, Bristol Medical School, University of Bristol, Bristol, UK
[2]Bristol Randomised Trials Collaboration, Population Health Sciences, University of Bristol, Bristol, UK
[3]NIHR School for Primary Care Research, Centre for Primary Care and Health Services Research, University of Manchester, Manchester, UK
[4]Institute of Health and Wellbeing, University of Glasgow, Glasgow, UK
[5]Population Health Sciences Division, Medical Research Institute, University of Dundee, Dundee, UK

**Acknowledgements** We would like to thank the patients and practices who participated in the study, the patients and carers who contributed to the advisory group and the other members of the 3D study team (Cindy Mann, Polly Duncan, Keith Moffatt, Emma Moody, Imran Rafi, Becca Robinson, Ali Heawood, Bryar Kadir, John McLeod). We would also like to thank Professor Joanna Coast and Dr Padraig Dixon for valuable guidance. This article is an abbreviated version of the final report to the funders (Salisbury C, Man M-S, Chaplin K, Mann C, Bower P, Brookes S, et al. A patient-centred intervention to improve the management of multimorbidity in general practice: the 3D RCT. Health Serv Deliv Res 2019;7(5)).

**Contributors** CS, PB, BG, SB, SH and SWM conceived, designed and interpreted results from the 3D trial. SH and JT designed the economic evaluation. M-SM (trial manager), KC, BF, CG and VL contributed to local economic data collection and interpretation of results. DG (statistician) conducted the multiple imputation. JT carried out the analysis and wrote the paper, and all authors revised it for important intellectual content.

**Funding** This project was funded by the National Institute for Health Research (NIHR) Health Services and Delivery Research Programme (project number 12/130/15). This study was designed and conducted in collaboration with the Bristol Randomised Trials Collaboration (BRTC), a UKCRC Registered clinical trials unit (CTU) in receipt of NIHR CTU support funding. CS is partly supported by The NIHR Collaboration for Leadership in Applied Health Research and Care West (CLAHRC West) and by Bristol Clinical Commissioning Group.

**Disclaimer**  The views and opinions expressed in this report are those of the authors and do not necessarily reflect those of the NIHR, the NHS or the Department of Health.

**Competing interests**  CS reports grants from CLAHRC West and Bristol CCG outside the submitted work. No other competing interests are declared.

**Patient consent for publication**  Not required.

**Ethics approval**  A favourable ethical opinion was granted by the South-West (Frenchay) National Health Service (NHS) Research Ethics Committee (14/SW/0011).

**Provenance and peer review**  Not commissioned; externally peer reviewed.

**Data availability statement**  Data are available upon reasonable request. Requests for data sharing should be submitted to Professor C Salisbury (c.salisbury@bristol.ac.uk) for consideration. Access to anonymised data might be granted following review via the University of Bristol Research Data Storage Facility.

**ORCID iDs**
Joanna Thorn http://orcid.org/0000-0001-8962-2428
Mei-See Man http://orcid.org/0000-0003-4948-5670
Katherine Chaplin http://orcid.org/0000-0003-1261-9938
Chris Salisbury http://orcid.org/0000-0002-4378-3960

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
