## [Reviewer comments · BMJ Open]

ARTICLE DETAILS

TITLE (PROVISIONAL)	Cost-effectiveness of a patient-centred approach to managing multimorbidity in primary care: a pragmatic cluster randomised controlled trial
AUTHORS	Thorn, Joanna; Man, Mei-See; Chaplin, Katherine; Bower, Peter; Brookes, Sara; Gaunt, Daisy; Fitzpatrick, Bridie; Gardner, Caroline; Guthrie, Bruce; Hollinghurst, Sandra; Lee, Victoria; Mercer, Stewart W; Salisbury, Chris

VERSION 1 – REVIEW

REVIEWER	Mieke Rijken NIVEL (Netherlands institute for health services research), The Netherlands Department of Health and Social Care Management, University of Eastern Finland, Finland
REVIEW RETURNED	24-Mar-2019

GENERAL COMMENTS	This study is well performed; the results are rather disappointing but need to be published. In general, the results are in line with other studies showing the uncertainty in terms of cost-effectiveness of comprehensive person-centred care interventions for patients with multimorbidity or other high-need patient populations (e.g., frail older people). I agree with the authors that we need more economic evaluations of such interventions before drawing firm conclusions, but I also wish some more critical consideration of the QALY outcome measures used in such evaluations. EQ-5D-5L has been the norm for many years, but it may not be able to capture the real benefits of such interventions for people with multimorbidity or other target populations in need of both health and social care. A broader concept of quality of life or wellbeing might be considered; I would appreciate to read something about this in the Discussion section of the manuscript. I have some additional points that need to be addressed: • Introduction: The description of the 3D intervention is too short and fragmented. There is some more information in the Method section (p. 7, lines 3-9), which could be replaced to the Introduction section, but it still needs more explanation. For instance, about the “patient-centred care model” underlying the intervention, a summary of the intervention elements and process and a clear definition of its target population. I am aware that the 3D intervention had been described extensively elsewhere, but readers of this manuscript should be able to find a clear summary of the intervention (perhaps in a box) here in this manuscript.• Also explain where 3D stands for.
---

	 • Minor issue in the Introduction lines 9-10: “Multimorbidity is the norm for older adults [...]”. I often see this phrase, but the term “norm” should be avoided here. (It is not a neutral term for something that is common. For instance, consider its use in “norms and values”). Better use “is common” or something like that. • Methods: This section lacks structure. Please provide a clear structure: Design, Context and sample, Measures, Statistical analysis (with subheadings per analysis and perspective). • Also already explain here the three perspectives: which types of costs will be included in estimations from the three perspectives. I can read between the lines, but it is easy to provide a more structured overview of types of costs per perspective. • What I miss is some more information about the selection criteria and recruitment of patients here. Again, I know that the 3D intervention has been described elsewhere, but readers of this study should be able to read the paper without the need to search for missing information elsewhere. • Results: Related to my previous comment, I wish to read more information about the initial patient sample, the non-response and participants in the study, as this is necessary to get an impression of the external validity of the study. In the study protocol I read that GPs could apply several exclusion criteria when composing the study sample and I really want to know more about this and whether there was any selection bias in the sample. In fact, the number of participating patients with three or more conditions per practice in the final sample seems low (40 or 50?), so what happened in selecting and inviting patients? • I prefer not to read that something is higher or lower in the intervention group compared with the control group, when not significant. I believe it is better to state that something is “similar” in both groups (as far as possible) in such cases. Please check this in the abstract, the Results and Discussion section. • In the Discussion it is concluded (p. 12, lines 53-56) that the participants in the study were substantial users of health care. This is not surprising, but given that the authors also suggest that cost-effectiveness may have been better if the intervention would focus on those patients with the greatest needs (p. 14, line 49), I wish the authors to discuss the representativeness of their study sample in relation to the external validity of the study. • Is there something to say about practice variation in the way the intervention was implemented, also resulting in practice variation in costs? • Also, structure the Discussion better. One heading should be the implementation of the intervention: how was the intervention implemented in the practices? And to what extent could this explain practice variation in outcomes and costs? • I think it is an understatement to say that “costs were slightly higher” in the intervention arm when based on the imputed dataset, and “slightly lower” when only complete cases were analysed. The differences in costs are very big, and I miss here again a discussion about the external validity of the study results. • At page 12, lines 33-35, it is estimated that 3.7% of the adults in the trial practices were eligible for the intervention AND considered suitable by their GP. This seems to me a very low percentage, considering the high prevalence of multimorbidity (also, when
--	--

	operationalised as ≥ 3 chronic conditions) in the UK and other countries. Why is it so low?  • Please add a critical discussion of the QALY measure EQ-5D. • There is one heading in the Discussion: Strengths and limitations. Lines 48-51 at page 14 do not belong under this heading. Would it be possible to add a paragraph with directions for future research? • Conclusions: Write separate sentences and skip the last part of the (long) last sentence starting with “however, given the modest implication costs, the intervention might be justified on the basis of the improvements observed in participants’ experience of patient-centred care”. This is not based on this study and should be left out the conclusions here. (This is definitely not to say that I disagree with this sentence, but it cannot be a conclusion of this particular study.)
--	---

REVIEWER	Myriam Soto-Gordoa Mondragon Unibertsitatea
REVIEW RETURNED	17-Apr-2019

GENERAL COMMENTS	The study aims to determine the cost-effectiveness of a patient-centred intervention that includes a) improvement of continuity of care, b) reduces patient burden in accessing healthcare and c) increases patient involvement in decision-making. Even though the article is relevant considering the need to change the current organizational models, authors should consider some significant aspects. The success of an intervention does not rely only on the intervention itself, but also on its implementation. Authors stated: “anticipated reductions in appointment attendances were not achieved, partly because some patients attended 3D reviews in addition to, rather than instead of, their single-condition reviews as intended. Similarly, although cost savings could have been achieved through a reduction in prescriptions issued, this was not observed in the trial”. This may suggest that the intervention was not implemented as planned, and therefore, it is hard to determine whether the intervention itself is cost-effective (Soto-Gordoa M, de Manuel E, Fullaondo A, et al. Impact of stratification on the effectiveness of a comprehensive patient-centered strategy for multimorbid patients. Health Serv Res. 2019;54:466-473). Major Comments  • Abstract: Methodology is a bit misleading. You have performed different analysis from different perspectives, yet results are only related to the cost-utility analysis • There is a lack of consistence on the description of the intervention in the abstract, introduction and methods. At the beginning, it seems that the intervention is a 3-dimension intervention that includes a) improvement of continuity of care, b) reduces patient burden in accessing healthcare and c) increases patient involvement in decision-making, but it seems that in reality the intervention focused on the first by enhancing 6-monthly GP consultations. Also in the discussion, you mention some software template. • I suggest ending the introduction section with the objectives of the program rather than with the conclusions • Methods: You should include the study design at the methods part • Methods: You should include the eligibility criteria for the patients. This is important since the selection of the population that
--

	will benefit for this kind of intervention will affect the effectiveness of the program.  • Methods: It would be very useful if you describe the current (control) way of treating those patients • Methods: It is recommended that you state why you did several analysis (cost-utility and cost-consequence) and why you choose different perspectives. Cost-utility analyses pose several methodological limitations (Huter, K, Kocot, E, Kissimova-Skarbek, K, Dubas-Jakóbczyk, K, & Rothgang, H 2016, 'Economic evaluation of health promotion for older people-methodological problems and challenges', BMC health services research, vol. 16, no. 5, p. 328) • Methods: please clarify what you mean by "costs and outcomes occurring during the final three months of follow-up were discounted". • Results: Add a descriptive analysis of the sample • Results. In Table 3 we can see that mean costs of private care increased in the intervention arm (although results are not statistically significant). Is there any reason for this? Minor Comments  • Abstract: 3D intervention is presented in the objective. I would suggested explaining the basis of the intervention from the very beginning. • Abstract: In the conclusion section: "... these results should be considered in conjunction with evidence from the participants themselves about their experience of patient-centred care and with other outcomes to inform decision on implementation". Can you please specify a bit more which other outcomes refer to? • Introduction. "Multimorbidity is the norm for older adults (with a prevalence of up to98%)". It would be interesting if you can you please specify for which age range you refer to. • Methods. In the "Economic outcomes" section you stated how Quality of Care was obtained. I would suggest reconsidering the heading. • Productivity loss. Clarify whether you refer to patients or to the caregivers. • Results. Primary analysis: "QALYs over 15 months compared with participants in the usual care arm (95%CI -0.009 to 0.023). Mean is missing. • Table1. "Adjusted costs from the NHS/PSS perspective". What is it adjusted by? • Table 1. Please add CI for ICER • Appendix 1. What do HCA, OOH, DEXa, ECG, etc stand for (Table 1)?
--	--

REVIEWER	Hanna Gyllensten University of Gothenburg, Sweden
REVIEW RETURNED	23-May-2019

GENERAL COMMENTS	You have conducted an interesting study, and I believe it is very important to add economic evaluations to this type of studies of changes in healthcare provision. However, some work is needed to make it easier for readers to understand what was done and how to interpret your results. METHODS You have a lot of information to present, and I find it somewhat difficult to find my way in the methods section and I believe some additional information is needed. Some of this is probably already
--

	presented in other publications from the project, but needed also here so that readers do not need to search for it. For example, information about the questionnaires is currently scattered throughout the Methods section, which makes it difficult to understand how many questionnaires are really sent out, and it is difficult to find the information about questionnaire time points. Are all data collected at the same time points (health outcomes, resource use and information from carers)? Do, for example, carers respond to the same questionnaires as the patients themselves? What instruments did the carers respond to, also EQ-5D-5L? If that is the case (0 - 9 - 15 months), can you comment (e.g., in the discussion/limitations) on the quite long time period it was expected that participants should remember details about their previous resource use (some studies claim that for less important treatment you cannot assume these are remembered longer than a couple of weeks) and what potential efforts were made to help them remember. Also, how well do these three measures really cover the changes in health-related quality of life over a 15 months long time-period? Can you add your definition and description of what is patient-centred care? What is the value of this and to whom? Since you state that your intervention actually resulted in a more patient-centred care, would it be more interesting to analyze costs vs patient-centredness (instead of health-related quality of life which you had already found were not affected by the intervention)? Also, can you in some way indicate why “usual care” is not patient-centred? Your manuscript and abstract includes the terms economic evaluation, cost-effectiveness, cost-utility analysis, cost-consequences analysis, and net monetary benefit. For readers without a background in health economics I believe these terms and the difference between these analyses needs to be clarified. What do you mean by the term “societal productivity perspective”? Is this only the indirect costs/lost productivity? For me, societal perspective/costs would include all costs to all payers. Can you clarify (page 8, row 51) how you adjusted for inflation, and is there a reference for the chosen level/percentage? In page 9, row 11, you state that some of the data downloaded were “clearly wrong”. This is also mentioned in the Discussion. Can you state in the Methods what you did to identify such errors, and possibly state also in the Discussion why such errors could occur? Can you develop the description of your multiple imputation? How was the imputation model developed, which variables were included and/or imputed (only resource use?), what model definitions were used and did you need to use e.g., truncation or nearest neighbors-solutions to get useful results? Were the assumptions for conducting imputation met, or can you add something about problems with the imputation in the Discussion/limitations? For how many individuals did you need to conduct imputations for each imputed variable? Were your cost data and/or health outcomes normally distributed? Were e.g., assumptions for conducting linear regression met? If data were skewed, have you handled this e.g., during the calculation of confidence intervals? Page 12, row 39: Can you add something about your instructions to participating health professionals, e.g., pharmacists, in the description of the intervention (Methods)? What were the “computerized ‘rules’” mentioned in the protocol, and how were
--	---

these used? In the studies I have previously seen they pharmacists main work has not been to suggest stopping medications, but to adhere to guidelines and identify potentially harmful medications. Was there an expectation that pharmacist reviews would result in decreased prescriptions? Did pharmacists receive information about e.g., adverse symptoms from the nurse reviewers, to base such suggestions on?

I also think it would benefit your manuscript if you added a study flow diagram and a table presenting the demographics of included patients and of patients with complete data et cetera, to make it easier to understand your results and how these are affected by missing data, deaths and withdrawals. Were the study population similar between groups (did your randomization and clustering work out as it should) or were there biases that you should have taken into account during the analysis?

Row 28: You inform readers that the method was changed, from the proposed analysis plan, but can you also write what was the initial plan?

Page 9, row 47: Can you explain how your “randomization variable” solved the clustering-issue? I expect it depends on how your randomization variable looks?

Page 14, row 34: I believe the software needs to be mentioned earlier in the manuscript.

RESULTS

With regards to presentation and interpretation of results, you have not totally convinced me that you found a zero-difference between 3D and usual care, and this is partly because you never test this statistically, and partly because it is difficult to follow the many different results presented (more a question of telling the story).

Regarding your main analysis and sensitivity analysis, you present the main analysis with imputed data (ITT), while the cost-consequence analysis is conducted on what you call “available cases” and the sensitivity analysis is conducted on “complete cases”. This is confusing, both when reading and when trying to understand the consequences of each alternative result. Would it be possible to present all analyses for all these alternative case inclusions, and to add an explanation of how differences in these results should be interpreted in the Discussion?

The same applies to discounting; according to page 8, rows 52-55, you discounted costs and outcomes by 3.5%. However, later in the manuscript it is clear that you made sensitivity analysis for this. I would suggest that this is stated in the methods and that results with different discounting rates are also presented in a table.

Maybe you could add an e-table/e-supplement with these types of “extra results”, for those of us readers who wants to know.

While reading I have thought a lot about your findings that there was not a lot of change in health outcomes (although slightly positive results), and not a lot of change in costs (although slightly negative), which is a common situation in these types of studies.

Presenting such results to readers is also difficult from a pedagogic standpoint. Have you considered adding an ad hoc equivalence analysis?

With this type of “no effect” studies it is also of interest to know that/if there were no specific sub-groups (e.g., healthier, younger, or those with high ratings in your evaluation of patient-centredness) in which the intervention appears to have had an effect. Have you looked at that? And which costs changed by introducing 3D? Which cost components in your list of resources were directly affected by this intervention? How should we understand the changes in other cost components? If your

	intervention increased costs in one part of this table, then a “zero-sum” effect should mean that some other component had a higher cost in the usual care arm, I expect? What were the expected changes and did the results look as you expected? Was the only effect in costs that you identified really that some of the patients in the intervention received both 3D-reviews and usual care, but otherwise no changes were made? Do you have a suggestion why carers would have lower QALY after introducing 3D? Was that expected? Is this a chance finding or can you see in the process evaluation that there was something going on that caused this? I read in the protocol that the reason for increasing the data collection period was that it took the practices 3 months before they started implementing the 3D-method. I could not find this information in the manuscript. What would be the effect of this on your results? Additionally, if the practices had difficulties implementing 3D, and there was a delay in introducing this, it is even more interesting to know the time line. How much time did they have to make a difference to each patient, and what could actually be expected in the time provided? If there was a delay, how many of the patients had time to conduct two “reviews” during the study period (before month 15), and how many of them had the medication review already at their first such review? Because if the medication review was conducted only just before they quit the study, I suppose not much could have been influenced by that review? Going back to the introduction, rows 16-25: I get the impression that there is not really any point conducting this study, can you possibly talk a bit more for your study? A related issue is the description and discussion about your comparators; the 3D intervention vs usual care. It sounds in the discussion as if you are not really changing to 3D-care, but actually adding 3D on top of usual care. I would suggest that you, at least in the Discussion, add something about how costs would have been affected by actually transferring to the 3D-alternative. Potentially, you could even make an ad hoc estimation of costs among people who followed the protocol, and compare to controls following usual care. Can you explain the results for net monetary benefit vs your ICER, also to a non-health economist? I have not calculated net monetary benefits myself, but it sounds strange, intuitively, that it was only valued at GBP 10 even though the ICER was GBP 1,500 below the threshold of GBP 20,000. I would also suggest that you revise the sentence in page 12, row 29, because now it sounds as if because it is a positive net monetary benefit the society would not be willing to pay for it? CLARIFICATIONS Minor clarifications suggested to ease reading: Article summary/strengths and limitations: Could you develop the last point provided so that it is clearer for the readers what is the meaning of these “wide confidence intervals” (e.g., large variation in costs)? Tables and appendices: I prefer it if all tables can be read on their own, e.g., that the currency is provided and that all abbreviations are handled in the same way (right now some abbreviations are explained in the tables and some below tables). Table 3: How should I read this table? Is N those with a response to the questions about that specific component, or those with that type of resource use? Is the difference a comparison between those with that resource use, or between the groups?
--	---

	Appendix 1: Can you explain (e.g., for international readers) the different prescription charge-items and what you mean by “band”? What do you mean by stating that EQ-5D at 9 months in secondary outcome and 15 months primary outcome, in the protocol? Does this affect your results in this manuscript in any way? Can you add information about participants consent to participate? Since you are presenting data from an RCT, can you add a CONSORT statement checklist?
--	--

VERSION 1 – AUTHOR RESPONSE

Reviewer: 1

This study is well performed; the results are rather disappointing but need to be published. In general, the results are in line with other studies showing the uncertainty in terms of cost-effectiveness of comprehensive person-centred care interventions for patients with multimorbidity or other high-need patient populations (e.g., frail older people). I agree with the authors that we need more economic evaluations of such interventions before drawing firm conclusions, but I also wish some more critical consideration of the QALY outcome measures used in such evaluations. EQ-5D-5L has been the norm for many years, but it may not be able to capture the real benefits of such interventions for people with multimorbidity or other target populations in need of both health and social care. A broader concept of quality of life or wellbeing might be considered; I would appreciate to read something about this in the Discussion section of the manuscript.

A discussion of the use of the EQ-5D has been added to the Discussion section: “The EQ-5D instrument was used because it is recommended by NICE in the UK25, and there was little evidence on alternatives at the trial outset. However, it is difficult to pick up small changes in health-related quality of life arising from service reconfigurations, and it may be that a measure capturing outcomes beyond health would be more appropriate. In future work, researchers should consider including alternative economic outcome measures alongside the EQ-5D; for example, the ICECAP suite of capability wellbeing instruments⁴².”

I have some additional points that need to be addressed:

Introduction: The description of the 3D intervention is too short and fragmented. There is some more information in the Method section (p. 7, lines 3-9), which could be replaced to the Introduction section, but it still needs more explanation. For instance, about the “patient-centred care model” underlying the intervention, a summary of the intervention elements and process and a clear definition of its target population. I am aware that the 3D intervention had been described extensively elsewhere, but readers of this manuscript should be able to find a clear summary of the intervention (perhaps in a box) here in this manuscript.

The 3D intervention has now been described more fully in Box 1. The target population for the intervention was patients with multimorbidity; this has now been explicitly stated in the methods, and the conditions included are listed in Table 1.

Also explain where 3D stands for.

3D was not intended to be an abbreviation, it was just a name and used to allude to the concept of wholeness, while also acting as a mnemonic for ‘dimensions of health, depression and drugs’. This has been clarified in Box 1.

Minor issue in the Introduction lines 9-10: "Multimorbidity is the norm for older adults [...]". I often see this phrase, but the term "norm" should be avoided here. (It is not a neutral term for something that is common. For instance, consider its use in "norms and values"). Better use "is common" or something like that.

The wording has been adjusted as suggested.

Methods: This section lacks structure. Please provide a clear structure: Design, Context and sample, Measures, Statistical analysis (with subheadings per analysis and perspective).

Thank you for the suggestion. We have added the headings Design and Measures, and re-organised the analysis section. However, as this is an economic evaluation, the existing headings draw attention to the appropriate parts of the methods, and 'Economic analyses' is more appropriate terminology than 'Statistical analysis' in this context. The individual analyses amount to little more than a sentence each, so we prefer to retain them under the broad heading.

Also already explain here the three perspectives: which types of costs will be included in estimations from the three perspectives. I can read between the lines, but it is easy to provide a more structured overview of types of costs per perspective.

The three perspectives have been described in more detail in the Economic analyses section.

What I miss is some more information about the selection criteria and recruitment of patients here. Again, I know that the 3D intervention has been described elsewhere, but readers of this study should be able to read the paper without the need to search for missing information elsewhere.

The selection criteria and recruitment process are described in detail in the final report and the Lancet paper, which has now been explicitly signposted in the methods. "Eligible patients were aged 18 years or older, with at least three types of chronic condition. Patients were excluded if they had a life expectancy of less than 12 months, were at serious suicidal risk, were known to be leaving the practice within 12 months, were unable to complete questionnaires in English, were taking part in another health-care research project, lacked the capacity to give consent (in Scotland only, for legal reasons), or if their general practitioner deemed them unsuitable to be invited for other reasons. If more than 150 patients per practice were eligible, a random sample of 150 potential participants were selected, their names were screened by their primary care physicians to exclude patients they judged unsuitable for research, and the remaining patients were invited by post to participate." (Salisbury C, Man M-S, Bower P, et al. Management of multimorbidity using a patient-centred care model: a pragmatic cluster-randomised trial of the 3D approach. *The Lancet* 2018;392(10141):41-50.)

Results: Related to my previous comment, I wish to read more information about the initial patient sample, the non-response and participants in the study, as this is necessary to get an impression of the external validity of the study. In the study protocol I read that GPs could apply several exclusion criteria when composing the study sample and I really want to know more about this and whether there was any selection bias in the sample. In fact, the number of participating patients with three or more conditions per practice in the final sample seems low (40 or 50?), so what happened in selecting and inviting patients?

Details of the initial patient sample have now been given in Table 1.

I prefer not to read that something is higher or lower in the intervention group compared with the control group, when not significant. I believe it is better to state that something is "similar" in both

groups (as far as possible) in such cases. Please check this in the abstract, the Results and Discussion section.

Economic evaluation is conducted within a framework that is not based on significance, but on the joint distribution of costs and outcomes. The point estimates provide the best estimates of costs and benefits, and we believe it is therefore acceptable to discuss them in this manner in the context of this study in which all costs and outcomes were similar between groups: in each case, the lack of significance is highlighted to allow readers to contextualise the text. Some of the issues are discussed here: Briggs AH, O'Brien BJ. The death of cost-minimization analysis? *Health economics* 2001;10(2):179-84.

In the Discussion it is concluded (p. 12, lines 53-56) that the participants in the study were substantial users of health care. This is not surprising, but given that the authors also suggest that cost-effectiveness may have been better if the intervention would focus on those patients with the greatest needs (p. 14, line 49), I wish the authors to discuss the representativeness of their study sample in relation to the external validity of the study.

Patients recruited to the trial had similar age and sex characteristics and had similar chronic conditions to the 3132 patients who were invited but did not respond, except that fewer had severe mental health problems, dementia, depression or learning disability. We do not have data about the quality of life or other patient reported measures of health status in non-participants. The point we are making in the discussion is that although overall the participants had poor quality of life and were substantial users of health care, not all individuals with multimorbidity will necessarily benefit from a comprehensive review and it may be more effective to target those with the greatest needs.

Is there something to say about practice variation in the way the intervention was implemented, also resulting in practice variation in costs?

The following text has been added to the discussion: "There was substantial variation between practices in the extent to which they implemented the variation and this may have contributed to variation in costs in the trial arm. Three practices barely implemented the intervention at all because of practice or staffing problems unconnected with the intervention, and this will have diluted the potential effects of the intervention overall."

Also, structure the Discussion better. One heading should be the implementation of the intervention: how was the intervention implemented in the practices? And to what extent could this explain practice variation in outcomes and costs?

Headings have been added to the Discussion, which has been substantially re-structured to follow the BMJ guidelines.

I think it is an understatement to say that "costs were slightly higher" in the intervention arm when based on the imputed dataset, and "slightly lower" when only complete cases were analysed. The differences in costs are very big, and I miss here again a discussion about the external validity of the study results.

We based this statement on both the statistical significance of the cost differences and their size in relation to overall costs. Costs differences were non-significant and small compared with overall costs (less than 3%); we therefore feel the terminology used is appropriate.

At page 12, lines 33-35, it is estimated that 3.7% of the adults in the trial practices were eligible for the intervention AND considered suitable by their GP. This seems to me a very low percentage, considering the high prevalence of multimorbidity (also, when operationalised as ≥ 3 chronic conditions) in the UK and other countries. Why is it so low?

This is not a low percentage in context. The prevalence in older people is certainly higher, but a figure of 3.7% of all adults aged 18 or over is in fact slightly higher than we anticipated. The prevalence of multimorbidity depends very much on which conditions are counted, and the age-range of the population studied. We defined it in terms of the 17 major conditions included in the Quality and Outcomes Framework, which we grouped into 10 types of conditions (for example coalescing asthma and COPD into one category). The prevalence rate in the trial was slightly higher than we anticipated based on a large study of the epidemiology of multimorbidity in the UK Br J Gen Pract 2011; DOI: 10.3399/bjgp11X548929 which used similar definitions.

Please add a critical discussion of the QALY measure EQ-5D.

Consideration of the EQ-5D measure has been added to the Discussion as described above.

There is one heading in the Discussion: Strengths and limitations. Lines 48-51 at page 14 do not belong under this heading. Would it be possible to add a paragraph with directions for future research?

A Future Work section has been added.

Conclusions: Write separate sentences and skip the last part of the (long) last sentence starting with "however, given the modest implication costs, the intervention might be justified on the basis of the improvements observed in participants' experience of patient-centred care". This is not based on this study and should be left out the conclusions here. (This is definitely not to say that I disagree with this sentence, but it cannot be a conclusion of this particular study.)

The Conclusion has been truncated as suggested.

Reviewer: 2

The study aims to determine the cost-effectiveness of a patient-centred intervention that includes a) improvement of continuity of care, b) reduces patient burden in accessing healthcare and c) increases patient involvement in decision-making. Even though the article is relevant considering the need to change the current organizational models, authors should consider some significant aspects. The success of an intervention does not rely only on the intervention itself, but also on its implementation. Authors stated: "anticipated reductions in appointment attendances were not achieved, partly because some patients attended 3D reviews in addition to, rather than instead of, their single-condition reviews as intended. Similarly, although cost savings could have been achieved through a reduction in prescriptions issued, this was not observed in the trial". This may suggest that the intervention was not implemented as planned, and therefore, it is hard to determine whether the intervention itself is cost-effective (Soto-Gordoa M, de Manuel E, Fullaondo A, et al. Impact of stratification on the effectiveness of a comprehensive patient-centered strategy for multimorbid patients. Health Serv Res. 2019;54:466-473).

This is very true; however, in the context of a study in which there was considerable uncertainty around all costs and outcomes, we think it would be challenging to demonstrate cost-effectiveness unequivocally however well the implementation had occurred. Additionally, the study was a pragmatic trial, in which implementation forms part of the intervention being studied. Further work is considering implementation issues more explicitly.

Major Comments

Abstract: Methodology is a bit misleading. You have performed different analysis from different perspectives, yet results are only related to the cost-utility analysis

It would not be possible to present all results in the abstract and it is common in economic evaluations to present results of the primary analysis. Results for the cost-consequences analysis referred to in the abstract, and other secondary analyses, are described in full in the body.

There is a lack of consistency on the description of the intervention in the abstract, introduction and methods. At the beginning, it seems that the intervention is a 3-dimension intervention that includes a) improvement of continuity of care, b) reduces patient burden in accessing healthcare and c) increases patient involvement in decision-making, but it seems that in reality the intervention focused on the first by enhancing 6-monthly GP consultations. Also in the discussion, you mention some software template.

The intervention has now been described in full in Box 1. Thank you for pointing out the oversight in not mentioning the template earlier - this has also been added to the Box.

I suggest ending the introduction section with the objectives of the program rather than with the conclusions

The objectives of the program have been expanded in the final paragraph of the Introduction. However, as this study is the economic evaluation alongside the trial, and the results of the trial have already been published, the results of the trial (but not the economic evaluation) need to be briefly discussed in the introduction; they provide the starting point for this further piece of work.

Methods: You should include the study design at the methods part

The study design has been expanded in the methods section.

Methods: You should include the eligibility criteria for the patients. This is important since the selection of the population that will benefit for this kind of intervention will affect the effectiveness of the program.

This is described in detail in the final report and the Lancet paper, which has now been explicitly signposted in the methods. "Eligible patients were aged 18 years or older, with at least three types of chronic condition. Patients were excluded if they had a life expectancy of less than 12 months, were at serious suicidal risk, were known to be leaving the practice within 12 months, were unable to complete questionnaires in English, were taking part in another health-care research project, lacked the capacity to give consent (in Scotland only, for legal reasons), or if their general practitioner deemed them unsuitable to be invited for other reasons. If more than 150 patients per practice were eligible, a random sample of 150 potential participants were selected, their names were screened by their primary care physicians to exclude patients they judged unsuitable for research, and the remaining patients were invited by post to participate." (Salisbury C, Man M-S, Bower P, et al. Management of multimorbidity using a patient-centred care model: a pragmatic cluster-randomised trial of the 3D approach. *The Lancet* 2018;392(10141):41-50.)

Methods: It would be very useful if you describe the current (control) way of treating those patients

This is now included in Box 1.

Methods: It is recommended that you state why you did several analysis (cost-utility and cost-consequence) and why you choose different perspectives. Cost-utility analyses pose several methodological limitations (Huter, K, Kocot, E, Kissimova-Skarbek, K, Dubas-Jakóbczyk, K, & Rothgang, H 2016, 'Economic evaluation of health promotion for older people-methodological problems and challenges', *BMC health services research*, vol. 16, no. 5, p. 328)

Thank you for the reference – that is an interesting paper. It is common in economic evaluations to conduct analyses from a variety of perspectives; as there is no one ‘right’ answer, this allows decision makers to consider a range of potential outcomes. In the UK, NICE require that analyses are conducted from the NHS and personal social services perspective; however, the broader societal perspective also has relevance (and is mandated in some jurisdictions). The justification for the societal perspective has been added to the methods section.

Methods: please clarify what you mean by “costs and outcomes occurring during the final three months of follow-up were discounted”.

To account for time preference (i.e. the idea that society tends to prefer having benefits now while paying costs in the future), economic evaluations apply an annual ‘discount’ rate to any costs/outcomes that occur after the first year of follow-up. As the follow-up period was 15 months in 3D, we discounted during the final three months.

Results: Add a descriptive analysis of the sample

The baseline characteristics of the sample are now given in a new Table 1.

Results. In Table 3 we can see that mean costs of private care increased in the intervention arm (although results are not statistically significant). Is there any reason for this?

We cannot think of any reason for this and assume it is due to chance. The difference is small, with wide confidence intervals which overlap.

Minor Comments

Abstract: 3D intervention is presented in the objective. I would suggested explaining the basis of the intervention from the very beginning.

The 3D intervention is now described in Box 1.

Abstract: In the conclusion section: “... these results should be considered in conjunction with evidence from the participants themselves about their experience of patient-centred care and with other outcomes to inform decision on implementation”. Can you please specify a bit more which other outcomes refer to?

As another reviewer suggested removing that part of the conclusion from the main body, we have taken it out. (However, the other outcomes might include process evaluation).

Introduction. “Multimorbidity is the norm for older adults (with a prevalence of up to 98%)”. It would be interesting if you can you please specify for which age range you refer to.

This figure came from a systematic review, which cited M. Fortin, G. Bravo, C. Hudon, A. Vanasse, L. Lapointe ‘Prevalence of multimorbidity among adults seen in family practice’ *Ann. Fam. Med.*, 3 (2005), pp. 223-228; it was referring to adults aged 65+.

Methods. In the “Economic outcomes” section you stated how Quality of Care was obtained. I would suggest reconsidering the heading.

The section covers quality-adjusted life years, which are commonly used as an economic outcome. Quality of care is not discussed.

Productivity loss. Clarify whether you refer to patients or to the caregivers.

Productivity losses refer to both patients and carers; this has now been clarified in the text in the Valuation of resource use section.

Results. Primary analysis: “QALYs over 15 months compared with participants in the usual care arm (95%CI -0.009 to 0.023). Mean is missing.

The mean difference was specified earlier in the sentence; the confidence interval has now been moved to sit alongside to make this clearer.

Table1. “Adjusted costs from the NHS/PSS perspective”. What is it adjusted by?

The estimates were adjusted for the randomisation variables (area, practice deprivation and practice size). This has been added to the bottom of the table.

Table 1. Please add CI for ICER

As the ICER is a ratio statistic, there are issues associated with the interpretation of confidence intervals; it is therefore recommended that the uncertainty be evaluated using other methods (e.g. cost-effectiveness acceptability curves), as is reported here.

Appendix 1. What do HCA, OOH, DEXa, ECG, etc stand for (Table 1)?

All abbreviations have now been listed at the bottom of the table.

Reviewer: 3

You have conducted an interesting study, and I believe it is very important to add economic evaluations to this type of studies of changes in healthcare provision. However, some work is needed to make it easier for readers to understand what was done and how to interpret your results.

METHODS

You have a lot of information to present, and I find it somewhat difficult to find my way in the methods section and I believe some additional information is needed. Some of this is probably already presented in other publications from the project, but needed also here so that readers do not need to search for it. For example, information about the questionnaires is currently scattered throughout the Methods section, which makes it difficult to understand how many questionnaires are really sent out, and it is difficult to find the information about questionnaire time points. Are all data collected at the same time points (health outcomes, resource use and information from carers)? Do, for example, carers respond to the same questionnaires as the patients themselves? What instruments did the carers respond to, also EQ-5D-5L?

Additional headings have been added to try to improve the navigation. The timepoints for data collection for the resource use using the patient-reported questionnaires have been described more coherently, and were the same as for outcome measurement. Carers responded to the EQ-5D-5L questionnaire at the same timepoints as the patients; this is now explicitly mentioned in the Outcome measurement section.

If that is the case (0 - 9 - 15 months), can you comment (e.g., in the discussion/limitations) on the quite long time period it was expected that participants should remember details about their previous resource use (some studies claim that for less important treatment you cannot assume these are remembered longer than a couple of weeks) and what potential efforts were made to help them remember. Also, how well do these three measures really cover the changes in health-related quality of life over a 15 months long time-period?

The evidence around recall period is quite varied, and there is no consensus on what the best recall period is. The most important resources (such as GP appointments, medications) were extracted from medical records. Recall period is now referred to in the discussion.

Issues surrounding the use of quality of life measures are now discussed in the discussion.

Can you add your definition and description of what is patient-centred care? What is the value of this and to whom? Since you state that your intervention actually resulted in a more patient-centred care, would it be more interesting to analyze costs vs patient-centredness (instead of health-related quality of life which you had already found were not affected by the intervention)? Also, can you in some way indicate why “usual care” is not patient-centred?

A description of patient-centred care has been added to the paper “Based on the seminal work of Stewart et al [ref], the concept of patient-centred care includes a focus on the patient’s individual illness experience, an integrated biopsychosocial perspective, seeking to find common ground with the patient and agreeing management plans, and enhancing a continuing relationship between the patient and doctor. We sought to apply these principles within the 3D intervention.”

There is evidence that this approach is valued by patients and that it is associated with improved health outcomes. Several recent guidelines e.g from the American Geriatric Society and from NICE have advocated a more patient-centred approach to multimorbidity. While it would have been interesting to use the measure of patient-centredness in the economic evaluation, this would not have enabled broad comparisons to be made with other interventions and did not form the base case described in our analysis plan.

Usual care is not patient-centred because chronic disease management has become increasingly siloed and delivered by different members of staff working to protocols which emphasise meeting the requirements of the Quality and Outcomes Framework. These mainly relate to undertaking specific processes and achieving specific outcomes in a standardised way. This approach does not pay attention to patients’ individual priorities and needs, tends to focus on biomedical aspects of care while ignoring quality of life, does not encourage sharing information and decisions with patients, and undermines continuity of care.

Your manuscript and abstract includes the terms economic evaluation, cost-effectiveness, cost-utility analysis, cost-consequences analysis, and net monetary benefit. For readers without a background in health economics I believe these terms and the difference between these analyses needs to be clarified.

We have now given a brief glossary as an appendix to the paper.

What do you mean by the term “societal productivity perspective”? Is this only the indirect costs/lost productivity? For me, societal perspective/costs would include all costs to all payers.

A societal perspective should indeed include all costs to all payers. However, it would have been impracticable to have collected all the data, particularly in this population of high users. We therefore opted to examine the societal contributions that we thought might have an impact, and used the term ‘societal productivity perspective’ to make it clear that only the productivity was included.

Can you clarify (page 8, row 51) how you adjusted for inflation, and is there a reference for the chosen level/percentage?

Apologies, the reference was accidentally omitted – it has now been added in.

In page 9, row 11, you state that some of the data downloaded were “clearly wrong”. This is also mentioned in the Discussion. Can you state in the Methods what you did to identify such errors, and possibly state also in the Discussion why such errors could occur?

The data were visually inspected; we have added this to the methods. We do not know of any reason why the errors might have occurred; it would be useful to conduct future work to identify the reason, and this has been added to a ‘Further work’ section.

Can you develop the description of your multiple imputation? How was the imputation model developed, which variables were included and/or imputed (only resource use?), what model definitions were used and did you need to use e.g., truncation or nearest neighbors-solutions to get useful results? Were the assumptions for conducting imputation met, or can you add something about problems with the imputation in the Discussion/limitations? For how many individuals did you need to conduct imputations for each imputed variable?

The multiple imputation was carried out by the statistics team for the main effectiveness analysis, and is described in the final report to the funders; space limitations preclude reporting this information in full here, but we have added a reference to where this full information can be found for the interested reader.

Were your cost data and/or health outcomes normally distributed? Were e.g., assumptions for conducting linear regression met? If data were skewed, have you handled this e.g., during the calculation of confidence intervals?

We are interested in the difference mean costs and QALYs between arms; it is generally safe to assume with a large enough sample size (such as that in 3D) that the differences between arms are approximately normally distributed even if the original source distributions were not normal.

Page 12, row 39: Can you add something about your instructions to participating health professionals, e.g., pharmacists, in the description of the intervention (Methods)? What were the “computerized ‘rules’” mentioned in the protocol, and how were these used? In the studies I have previously seen they pharmacists main work has not been to suggest stopping medications, but to adhere to guidelines and identify potentially harmful medications. Was there an expectation that pharmacists reviews would result in decreased prescriptions? Did pharmacists receive information about e.g., adverse symptoms from the nurse reviewers, to base such suggestions on?

Pharmacists were briefed to optimise medication. They were asked to identify non-essential drugs that could be stopped, essential drugs that should be started, and seek ways to simplify drug treatment regimes eg by making all doses once daily. The nurse, pharmacist and GP all used the same computerised template which encouraged them to work in a more patient-centred way. For example the first item on the template was to ask the patient what aspect of their health was their biggest problem. The template followed the same structure for every patient but the specific questions that appeared depended on the patient’s specific health conditions. The template also incorporated various checks based on national guidelines, for example warning GPs if a patient with heart disease was not taking a statin. The nurse, pharmacist and GP completed their own sections of the template but they could see what the others had written, so it acted as a shared record, and the final report given to patients included mail-merged information from the different sections. The pharmacist instructions have been clarified in Box 1.

I also think it would benefit your manuscript if you added a study flow diagram and a table presenting the demographics of included patients and of patients with complete data et cetera, to make it easier to understand your results and how these are affected by missing data, deaths and withdrawals. Were

the study population similar between groups (did your randomization and clustering work out as it should) or were there biases that you should have taken into account during the analysis?

The baseline characteristics of the patients and the practices are now given in Table 1, and the flow diagram of the study is given in Figure 1.

Row 28: You inform readers that the method was changed, from the proposed analysis plan, but can you also write what was the initial plan?

The original plan was to use the EQ-5D-5L value set for England (Devlin, N., et al., Valuing health-related quality of life: an EQ-5D-5L value set for England. 2016). However, NICE issued a position statement advising the use of the Van Hout crosswalk method instead.

Page 9, row 47: Can you explain how your “randomization variable” solved the clustering-issue? I expect it depends on how your randomization variable looks?

Each practice represented a cluster, randomised by area (Bristol, Manchester, Glasgow), practice deprivation and practice size. These randomisation variables were included in the regression to obtain the cost and QALY estimates. This information has been added to the tables.

Page 14, row 34: I believe the software needs to be mentioned earlier in the manuscript.

The software (an interactive computerised template) is now mentioned in the Box describing the intervention.

RESULTS

With regards to presentation and interpretation of results, you have not totally convinced me that you found a zero-difference between 3D and usual care, and this is partly because you never test this statistically, and partly because it is difficult to follow the many different results presented (more a question of telling the story).

Regarding your main analysis and sensitivity analysis, you present the main analysis with imputed data (ITT), while the cost-consequence analysis is conducted on what you call “available cases” and the sensitivity analysis is conducted on “complete cases”. This is confusing, both when reading and when trying to understand the consequences of each alternative result. Would it be possible to present all analyses for all these alternative case inclusions, and to add an explanation of how differences in these results should be interpreted in the Discussion?

The main trial results indicate that there was no difference between 3D and usual care in terms of the main outcome (Salisbury C, Man M-S, Bower P, et al. Management of multimorbidity using a patient-centred care model: a pragmatic cluster-randomised trial of the 3D approach. *The Lancet* 2018;392(10141):41-50.). The main cost-effectiveness analysis was conducted with all participants as randomised, and the complete-case analysis was simply a sensitivity analysis. The cost-consequences analysis has a different purpose, and was therefore conducted using as much data as we had available. It is quite common in economic evaluations to conduct multiple analyses using different groups of patients, as each one contributes information to an overall picture. The Discussion covers the reasons for the complete case analysis differing.

The same applies to discounting; according to page 8, rows 52-55, you discounted costs and outcomes by 3.5%. However, later in the manuscript it is clear that you made sensitivity analysis for this. I would suggest that this is stated in the methods and that results with different discounting rates

are also presented in a table. Maybe you could add an e-table/e-supplement with these types of “extra results”, for those of us readers who wants to know.

The primary analyses included discounting, while the sensitivity analysis did not include discounting (ie a discount rate of zero); we have adjusted the text to try to make this clearer. The results were trivially different, as might be expected from a 3-month period.

While reading I have thought a lot about your findings that there was not a lot of change in health outcomes (although slightly positive results), and not a lot of change in costs (although slightly negative), which is a common situation in these types of studies. Presenting such results to readers is also difficult from a pedagogic standpoint. Have you considered adding an ad hoc equivalence analysis?

Thank you for the suggestion. However, as the analysis was guided by the use of a health economics analysis plan, we would prefer not to deviate from this without very strong reasons. It would certainly be worth considering the inclusion of equivalence analyses in future studies, and we will bear that in mind.

With this type of “no effect” studies it is also of interest to know that/if there were no specific sub-groups (e.g., healthier, younger, or those with high ratings in your evaluation of patient-centredness) in which the intervention appears to have had an effect. Have you looked at that? And which costs changed by introducing 3D? Which cost components in your list of resources were directly affected by this intervention? How should we understand the changes in other cost components? If your intervention increased costs in one part of this table, then a “zero-sum” effect should mean that some other component had a higher cost in the usual care arm, I expect? What were the expected changes and did the results look as you expected? Was the only effect in costs that you identified really that some of the patients in the intervention received both 3D-reviews and usual care, but otherwise no changes were made? Do you have a suggestion why carers would have lower QALY after introducing 3D? Was that expected? Is this a chance finding or can you see in the process evaluation that there was something going on that caused this?

Given the substantial uncertainty around all cost estimates, we cannot read anything into the changes in cost components, or reliably describe them as a ‘zero-sum’ effect. The analysis was guided by a pre-specified analysis plan, and we would prefer not to deviate from this without very strong reasons. Carer issues are being considered in further work.

I read in the protocol that the reason for increasing the data collection period was that it took the practices 3 months before they started implementing the 3D-method. I could not find this information in the manuscript. What would be the effect of this on your results? Additionally, if the practices had difficulties implementing 3D, and there was a delay in introducing this, it is even more interesting to know the time line. How much time did they have to make a difference to each patient, and what could actually be expected in the time provided? If there was a delay, how many of the patients had time to conduct two “reviews” during the study period (before month 15), and how many of them had the medication review already at their first such review? Because if the medication review was conducted only just before they quit the study, I suppose not much could have been influenced by that review?

The 3 month delay was mainly to do with trying to do a randomised trial of a complex service intervention, rather than difficulties experienced by practices with the intervention. After practices had been randomised, we had to train all the staff, install the software, change practices recall systems, and then invite patients to their first 3D appointment giving them a reasonable amount of notice of their appointment. In our piloting we observed it took about three months between randomisation and when the first patient had their 3D review. Hence the change to a 15 month follow-up time point, to ensure that most participants had the

opportunity to have their first review before the first follow-up time point. The medication review was only conducted once, usually with the first review before 9 months, so well before the end of the trial after 15 months. 76% (607/797) had a pharmacist medication review. 75% (595/797) of patients had at least one full multidisciplinary 3D review and 390/797 (49%) of patients had two full multidisciplinary reviews in 15 months.

Going back to the introduction, rows 16-25: I get the impression that there is not really any point conducting this study, can you possibly talk a bit more for your study? A related issue is the description and discussion about your comparators; the 3D intervention vs usual care. It sounds in the discussion as if you are not really changing to 3D-care, but actually adding 3D on top of usual care. I would suggest that you, at least in the Discussion, add something about how costs would have been affected by actually transferring to the 3D-alternative. Potentially, you could even make an ad hoc estimation of costs among people who followed the protocol, and compare to controls following usual care.

The need for studies such as the 3D trial was expressed in several of the reviews of the evidence. The inconsistency in the evidence for cost-effectiveness is a clear indication that a large, well-conducted trial needed to be carried out, and the NIHR was prepared to fund a study to achieve this aim. Conducting this study has added to the information about how this type of intervention might be interpreted in practice; for example, we found that some practices used the 3D reviews in addition to the individual condition reviews. The possible effect on cost-effectiveness has been made clearer in the Discussion.

Can you explain the results for net monetary benefit vs your ICER, also to a non-health economist? I have not calculated net monetary benefits myself, but it sounds strange, intuitively, that it was only valued at GBP 10 even though the ICER was GBP 1,500 below the threshold of GBP 20,000. I would also suggest that you revise the sentence in page 12, row 29, because now it sounds as if because it is a positive net monetary benefit the society would not be willing to pay for it?

The net monetary benefit statistic is more useful, because it is more stable to small changes in outcome. As a ratio statistic, the ICER is very sensitive to small QALY differences, which is why it appears so much 'better' in this case. The identified sentence has been changed to read "The net monetary benefit was small, but positive, indicating that society is willing to pay for the benefits that can be achieved."

CLARIFICATIONS

Minor clarifications suggested to ease reading:

Article summary/strengths and limitations: Could you develop the last point provided so that it is clearer for the readers what is the meaning of these "wide confidence intervals" (e.g., large variation in costs)?

An explanation has been added "This leads to wide confidence intervals that cross zero (i.e. substantial uncertainty) for estimates of cost-effectiveness and illustrates the difficulty of obtaining precise estimates for the economic outcomes of this type of intervention even in large trials."

Tables and appendices: I prefer it if all tables can be read on their own, e.g., that the currency is provided and that all abbreviations are handled in the same way (right now some abbreviations are explained in the tables and some below tables).

Currency has been restored to Table 2 and abbreviations are now listed at the bottom of the tables.

Table 3: How should I read this table? Is N those with a response to the questions about that specific component, or those with that type of resource use? Is the difference a comparison between those with that resource use, or between the groups?

N represents those for whom we have data on that question (through a questionnaire, practice downloads, or medical records). The difference is the comparison of the groups.

Appendix 1: Can you explain (e.g., for international readers) the different prescription charge-items and what you mean by "band"?

The following explanatory text has been added to the foot of the table: "** Where prescription charges are levied, patients can choose to pay for each individual item separately, or can purchase a pre-payment certificate to cover any number of prescriptions in a 3- or 12-month period ** NHS bands refer to payscales: for example, a recently qualified nurse will be paid on band 5, while a nurse with more experience might be paid on band 6."

What do you mean by stating that EQ-5D at 9 months in secondary outcome and 15 months primary outcome, in the protocol? Does this affect your results in this manuscript in any way?

We defined the primary outcome as the EQ5D at 15months and treated the EQ5D at 9 months as a secondary outcome for the clinical effectiveness analysis. However for the economic analysis, we calculated QALYs which use the data from both time points. The results in this paper are based on QALYs as intended from the start, and in common with many economic evaluations.

Can you add information about participants consent to participate?

Patients with 3 or more conditions were mailed information about the study and a consent form, along with the baseline questionnaire. Only those returning signed consent were included.

Since you are presenting data from an RCT, can you add a CONSORT statement checklist?

As this is an economic evaluation, the appropriate checklist is the CHEERS statement checklist, which was supplied with the original submission.

VERSION 2 – REVIEW

REVIEWER	Hanna Gyllensten Institute of Health and Care Sciences, University of Gothenburg, Sweden
REVIEW RETURNED	25-Jul-2019

GENERAL COMMENTS	Thank you for your careful revisions. I have only a few minor comments to add: 1. I would suggest that you mention the use of informed consent somewhere in the manuscript. 2. It is still in some parts of the manuscript unclear to me if the "societal productivity perspective" includes all of the measured costs or if it is only productivity costs. This should be clarified. 3. Regarding "talking for your study" in the Introduction; I still believe you could benefit from stating why this study was needed. Why is the 3D trial a response to the "Several reviews have
---

	highlighted the need for research on new interventions"? You write about it in your response, but not in the manuscript. 4. Is reference 13 correctly assigned to the information about multiple imputation? 5. Where do you provide the reference to the full imputations model information? I did not find this now. Moreover, I am used to having to add all relevant checklists, thus often adding 2-3 different (e.g., CHEERS and STROBE checklists) to each publication. Maybe the editorial office can provide guidance?
--	---

VERSION 2 – AUTHOR RESPONSE

Reviewer: 3

Reviewer Name: Hanna Gyllensten

1. I would suggest that you mention the use of informed consent somewhere in the manuscript.

A reference to informed consent has now been added to the Methods section under 'Design: The 3D trial':-

"and all patients and practices gave written informed consent to participate in the study."

2. It is still in some parts of the manuscript unclear to me if the "societal productivity perspective" includes all of the measured costs or if it is only productivity costs. This should be clarified.

The primary analysis was from the NHS/PSS perspective. Productivity losses were only included in the secondary cost-consequences analysis, and were therefore reported in disaggregated form (ie the societal productivity perspective only includes productivity losses). If we had conducted the primary analysis from the societal perspective, we would have included all the measured costs.

As the term is causing confusion, we have dropped the word 'societal' to clarify that we are referring to productivity losses in this particular case.

3. Regarding "talking for your study" in the Introduction; I still believe you could benefit from stating why this study was needed. Why is the 3D trial a response to the "Several reviews have highlighted the need for research on new interventions"? You write about it in your response, but not in the manuscript.

The following text has now been added to the introduction to justify the trial: "The inconsistency in the evidence for cost-effectiveness provided a clear indication that a large, well-conducted trial with an integral economic evaluation needed to be carried out."

4. Is reference 13 correctly assigned to the information about multiple imputation?

Although reference 13 does contain some information about the multiple imputation, the full details are given in reference 14. This has now been updated where appropriate – many thanks for picking it up.

5. Where do you provide the reference to the full imputations model information? I did not find this now.

The full imputation model information is given on page 28 of reference 14 (Salisbury C, Man M-S, Chaplin K, Mann C, Bower P, Brookes S, et al. A patient-centred intervention to improve the management of multimorbidity in general practice: the 3D RCT. Health Serv Deliv Res 2019;7(5). This reference has now been cited correctly at the appropriate point.

Moreover, I am used to having to add all relevant checklists, thus often adding 2-3 different (e.g., CHEERS and STROBE checklists) to each publication. Maybe the editorial office can provide guidance?

The CONSORT checklist was appropriately submitted alongside the clinical effectiveness paper, and would not be relevant to a standalone economic evaluation. Furthermore, I have not seen any evidence that standalone economic evaluations published in BMJ Open have been accompanied by CONSORT statements. The CHEERS checklist has been submitted for this paper.